# The microRNA-mediated gene regulatory network in the hippocampus and hypothalamus of the aging mouse

Choijamts Munkhzul[1,2], Sun Shin Yi[3], Junhyung Kim[1,2], Seongsoo Lee[4,5], Hyuntae Kim[4], Jong-Seok Moon[1,2]*, Mihye Lee[1,2]*

**1** Soonchunhyang Institute of Medi-Bio Science, Soonchunhyang University, Cheonan, Korea, **2** Department of Integrated Biomedical Science, Soonchunhyang University, Cheonan, Korea, **3** Department of Biomedical Laboratory Science, College of Medical Sciences, Soonchunhyang University, Asan, Korea, **4** Gwangju Center, Korea Basic Science Institute (KBSI), Gwangju, Korea, **5** Department of Systems Biotechnology, Chung-Ang University, Anseong, Korea

* mihyelee@sch.ac.kr (ML); jongseok81@sch.ac.kr (JSM)

**Data Availability Statement:** RNA sequencing data generated in this study are available in the Gene Expression Omnibus (GEO) under GSE231582.

## Abstract

Aging leads to time-dependent functional decline of all major organs. In particular, the aging brain is prone to cognitive decline and several neurodegenerative diseases. Various studies have attempted to understand the aging process and underlying molecular mechanisms by monitoring changes in gene expression in the aging mouse brain using high-throughput sequencing techniques. However, the effect of microRNA (miRNA) on the post-transcriptional regulation of gene expression has not yet been comprehensively investigated. In this study, we performed global analysis of mRNA and miRNA expression simultaneously in the hypothalamus and hippocampus of young and aged mice. We identified aging-dependent differentially expressed genes, most of which were specific either to the hypothalamus or hippocampus. However, genes related to immune response-related pathways were enriched in upregulated differentially expressed genes, whereas genes related to metabolism-related pathways were enriched in downregulated differentially expressed genes in both regions of the aging brain. Furthermore, we identified many differentially expressed miRNAs, including three that were upregulated and three that were downregulated in both the hypothalamus and hippocampus. The two downregulated miRNAs, miR-322-3p, miR-542-3p, and the upregulated protein-encoding coding gene *C4b* form a regulatory network involved in complement and coagulation cascade pathways in the hypothalamus and hippocampus of the aging brain. These results advance our understanding of the miRNA-mediated gene regulatory network and its influence on signaling pathways in the hypothalamus and hippocampus of the aging mouse brain.

## Introduction

Aging is an irreversible, time-dependent process marked by the decline of physiological functions responsible for survival and fertility [1]. This physiological decline leads to loss of organ

Additionally, the minimal data sets for Figs 4C, 4E, S4D, and S4E and minimal data set for Fig 4D are available as Supporting Information files.

**Funding:** This work was supported by the National Research Foundation of Korea (Grant NRF-2021R1A2C4002421 to M.L.; RS-2023-00219563 to M.L.; 2021R1C1C1007810 to J.M.) and Soonchunhyang University Research Fund (Soonchunhyang University Research Fund 2021 to J.M.). The funders had no role in study design, data collection and analysis, decision to publish, or preparation of the manuscript.

**Competing interests:** The authors have declared that no competing interests exist.

and tissue function and an increased risk of cancer, diabetes, and cardiovascular disease [2]. Aging of the brain is often associated with a decline in motor, sensory, and cognitive functions, resulting in common neurodegenerative diseases such as Alzheimer's and Parkinson's disease [3]. To understand the aging processes, the molecular mechanisms of aging, as well as morphological and physiological changes in the aging brain, have been extensively investigated [4]. In particular, analysis of gene expression changes associated with brain aging can characterize cellular events during aging and may allow prevention or treatment of age-related diseases [5]. Gene expression changes during aging have been reported at cellular, tissue and organism level using *Drosophila* [6, 7], *Caenorhabditis elegans* [8], and mouse models [9]. Several studies have identified differentially expressed genes (DEGs) by comparing young and aged mice, mostly focusing on specific brain regions [10]. Using RNA sequencing (RNA-seq), Shavlakadze et al. [11] identified 229 age-regulated genes with a significant proportion implicated in immune-response signaling pathways in the hippocampal region of rats. Li et al. [12] performed RNA-seq of the hippocampal regions of the mouse brain during aging and found DEGs associated with neuroinflammation. Ximerakis et al. [13] performed single-cell RNA-sequencing (scRNA-seq) of whole brains of young and aged mice and found DEGs involved in protein synthesis, oxidative stress, inflammatory responses, and growth factor signaling. However, the underlying control mechanisms of genome-wide gene expression in aging are still not comprehensively understood.

miRNAs are small non-coding RNAs composed of 19–24 nucleotides that regulate gene expression at the post-transcriptional level [14, 15]. miRNA-mediated regulation of gene expression is critical for most biological processes, including tissue development, cellular differentiation, cell proliferation, and cell death [16]. In humans, 60% of protein-coding genes are regulated by miRNAs [17]. Numerous studies have shown that miRNAs play an important role in regulating aging processes, such as cognitive decline, inflammation, and neurodegenerative diseases, in mouse brains [18–20]. Li et al. [21] analyzed miRNA expression datasets from human prefrontal cortex obtained from individuals aged 2 days to 98 years and proposed that age-related miRNAs play an important role during development and the aging process of the human brain. Danka Mohammed et al. [22] performed small-RNA sequencing (small-RNA seq) of mouse brain hippocampal regions in young and aged mice and identified a differentially upregulated miR-204 target *Ephb2*, which may be involved in age-associated cognitive decline. Inukai et al. [23] investigated miRNA expression in whole brains of young (5-month-old) and aged (24–25-month-old) mice, and found that the predicted target genes of eight age-dependent upregulated miRNAs were enriched in the insulin signaling pathway, which plays a significant role in the aging process [24].

Our understanding of miRNA-mediated regulation of gene expression in the complex aging process remains incomplete. We therefore performed simultaneous analysis of mRNA and miRNA expression in the hypothalamic and hippocampal regions of young and aged mouse brains. Our sequencing data indicated early signatures of aging when comparing gene expression in young (2-month-old) and aged (15-month-old) samples. Several DEGs, including *C4b*, were confirmed in publicly available single-nucleus RNA sequencing (snRNA-seq) data from young (4-month-old) and aged (24-month-old) mouse brains. In addition, we identified differentially expressed miRNAs during aging and predicted the regulatory gene networks of miRNAs and their targets. The complement and coagulation cascades were notably associated with miRNA-mediated gene expression changes during brain aging in both the hypothalamus and hippocampus. Furthermore, coagulation factor III, a primary initiator of the extrinsic coagulation pathway, was detected at a higher level in aged mouse brain than in young mouse brain.

## Results

### Transcriptomic analysis of mRNA expression from the hippocampus and hypothalamus of aging mouse brains

Age-dependent gene expression changes in the mouse brain have been primarily examined in the late stages (24–27-month-old) of aging [11, 12]. Here, we focused on understanding the early events in brain aging, which could reveal triggering factors of the aging process. We investigated mRNA and miRNA expression changes in the relatively early aging process using young (2-month-old) and aged (15-month-old) C57BL/6J mice. Brain tissue has a complex structure with multiple regions [25, 26]. The hypothalamus is a crucial structure in the brain region involved in neuronal, endocrine, homeostatic, and aging processes [27–30]. Age-related changes are observed in the neurons of the hypothalamus that control metabolism and energy expenditure during obesity-associated aging [29, 31, 32]. The hippocampus plays an important role in regulating learning, encoding memory, and memory consolidation, and participates in cognitive aging [33–36]. Age-related decline in hippocampal volume has been well documented and correlates with Alzheimer's disease and mild cognitive impairment [37–39]. We therefore examined tissue-specific signatures in both the hypothalamus and hippocampus during brain aging. Total RNA isolated from hypothalamic and hippocampal tissues of young and aged mice were used for both mRNA and small RNA sequencing (Fig 1A and S1A Fig). At the transcriptome level, 17,997 genes were analyzed in hypothalamic tissue (S1 Table). Differential expression analysis revealed 74 downregulated genes and 65 upregulated genes (S1B Fig). A total of 17,818 genes were analyzed in hippocampal tissues (S1 Table), with 117 downregulated genes and 83 upregulated genes (fold change > 1.5, $p_{adj} < 0.05$) (S1C Fig). Among differentially expressed genes (DEGs), we found 20 upregulated and 34 downregulated genes common to both hypothalamic and hippocampal tissues (fold change > 1.5, $p_{adj} < 0.05$) (Fig 1B and 1C). Brain region-specific transcriptome analysis also revealed hypothalamus- or hippocampus-specific DEGs, including 45 upregulated and 40 downregulated hypothalamic genes, and 63 upregulated and 83 downregulated hippocampal genes (Fig 1B and 1C). Individual DEGs identified in our data have also been reported in previous studies. Pluvinage et al. [40] found that age-dependent upregulation of *Cd22* in hippocampal tissue is related to cognitive decline in old mice. Furthermore, interleukin-1 receptor-associated kinase 3 (*Irak3*), a cytokine signaling modulator that play an important role in obesity and metabolic syndromes by regulating inflammatory responses, is upregulated with age [41]. Li M et al. [12] found cognitive dysfunction related Defb1, Spag6, and Ly9 genes that are differentially downregulated in the hippocampal tissue of aged mice (12 months) when compared with young mice (2 months), and these genes are reportedly involved in apoptosis and inflammation during aging.

### KEGG enrichment analysis of differentially expressed genes

In our analysis, the expression level changes for most genes were under 1.5-fold, with the exception of 54 genes in both tissues, suggesting that changes in mRNA expression are not dramatic during brain aging. Although expression levels of individual genes are only slightly altered, the interconnected gene networks of DEGs can play a triggering role in the aging process and drive critical events. Therefore, to investigate biological pathways linked to DEGs, we performed Kyoto Encyclopedia of Genes and Genomes (KEGG) pathway analysis of upregulated and downregulated genes in hippocampal and hypothalamic tissues. For hypothalamic tissues, upregulated genes were significantly enriched in cytosolic DNA-sensing and B-cell receptor signaling pathways, whereas downregulated genes were enriched in pyrimidine metabolism and leukocyte transendothelial migration pathways (Fig 1D). For hippocampal

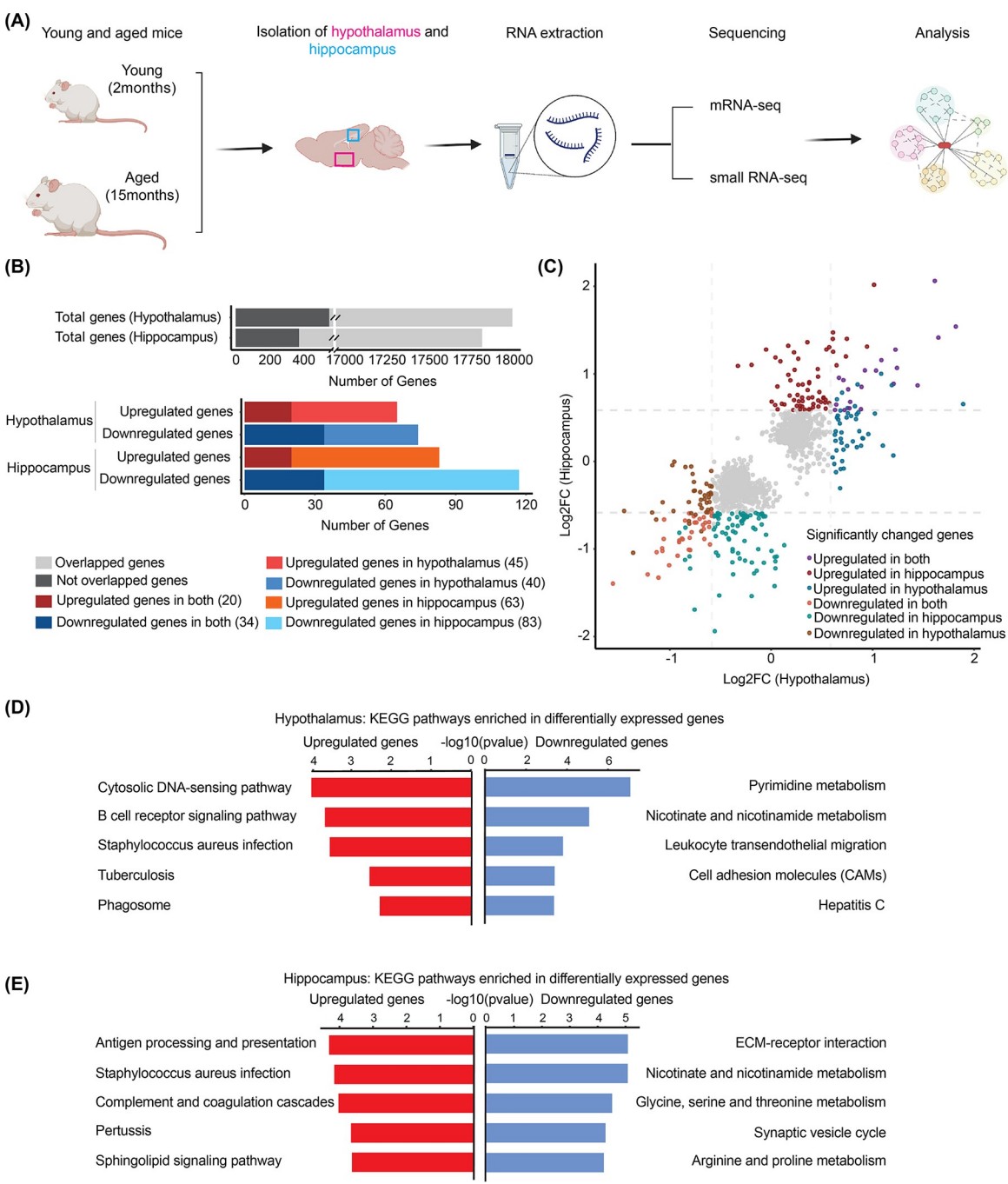

**Fig 1. Differentially expressed genes in the hypothalamus and hippocampus of young and aged mice.** (A) Schematic of the experimental design. Hypothalamic and hippocampal tissues were dissected from young (2-month-old) and aged (15-month-old) mice. Total RNA was isolated, followed by mRNA and small-RNA sequencing analysis. (Created with Biorender.com) (B) Bar graph showing the number of total detected genes and brain region-specific genes (dark gray) in hypothalamic or hippocampal tissue. These genes were further categorized as either upregulated or downregulated genes (cut off: fold change > 1.5, $p_{adj}$ < 0.05). (C) Scatter plot of expression level changes for a gene set detected in both hypothalamic and hippocampal tissues. Significantly upregulated or downregulated genes (fold change > 1.5, $p_{adj}$ < 0.05) in the hypothalamic or hippocampal tissues during aging are highlighted with colors. (D) Top five KEGG signaling pathways associated with differentially upregulated (red) and downregulated (blue) genes in hypothalamic tissue. (E) Top five KEGG signaling pathways associated with differentially upregulated (red) and downregulated (blue) genes in hippocampal tissue.

tissues, KEGG pathway enrichment analysis revealed that upregulated genes were enriched in complement and coagulation cascades, and antigen processing and presentation pathways, whereas downregulated genes were enriched in ECM-receptor interaction and glycine, serine, and threonine metabolic pathways (Fig 1E). Taken together, during brain aging, upregulated genes are enriched in immune-related pathways in both hypothalamic and hippocampal tissues, while downregulated genes are associated with metabolic pathways. The aging brain exhibits systemic inflammation and impaired immune responses and undergoes decreased energy utilization and increased mitochondrial dysfunction [42–52]. Hypometabolism and inflammation, which are closely connected, correlate with cognitive decline and increase the risk of age-related diseases, including neurodegenerative disease [42, 53]. Our results suggest that age-dependent gene expression changes underlie immune and metabolic system dysfunction, thus highlighting the relevance of gene regulatory networks in brain aging.

## Cell-type specific DEG analysis using single-nucleus RNA sequencing data

We next investigated from which cell types in the mouse brain the age-related signatures found in our analysis originated. We compared our bulk mRNA-seq data to publicly available snRNA-seq data from hypothalamic (GSE188646) [54] and hippocampal (GSE161340) [55] regions, although snRNA-seq only provides nuclear mRNA levels [56]. First, we verified the total coverage of gene expression level analysis in the bulk mRNA-seq and snRNA-seq. In hypothalamic tissue, our bulk RNA-seq identified 17,997 genes, of which 57.7% were detected in snRNA-seq data [54] (Fig 2A). In hippocampal tissue, our bulk mRNA-seq identified a total of 17,818 genes, of which 56% were found in snRNA-seq data [55] (Fig 2B). We also analyzed the number of overlapping genes for each cell type and found a higher number of overlapping genes in neuronal cell types (S2A and S2B Fig). We then calculated the module score of relative expression for the genes common to both bulk mRNA-seq and snRNA-seq data within the identified cell types [57]. In hypothalamic tissue, the module scores of astrocytes, microglia/macrophages, oligodendrocytes, tanycytes, and vascular leptomeningeal cells (VLMCs) were similar, whereas ependymocytes, oligodendrocyte precursors (OPCs), and pericytes/endothelial cells exhibited relatively higher scores. Neurons exhibited a distinctly high module score in the hypothalamus (Fig 2C). In hippocampal tissue, the module scores of neurons and interneurons were also higher than those of microglia, oligodendrocytes, astrocytes, and OPCs (Fig 2D). Brain cells have more neurons compared with other non-neuronal glial cells [58], which accounts for the high module scores in our analysis. We then evaluated DEG expression levels from our bulk mRNA-seq and snRNA-seq data. Among 65 upregulated and 74 downregulated genes from hypothalamic bulk mRNA-seq data, five upregulated and seven downregulated genes in astrocytes, seven upregulated and nine downregulated genes in neurons, and five upregulated and eight downregulated genes in oligodendrocytes were detected in snRNA-seq data (Fig 2E). In the hippocampus, 12 upregulated and 16 downregulated genes in astrocytes, 16 upregulated and 18 downregulated genes in neurons, and 14 upregulated and 16 downregulated genes in oligodendrocytes were found in snRNA-seq data among the total of 83 upregulated and 117 downregulated genes (Fig 2H). Only a small number of overlapping genes were detected in snRNA-seq data, probably due to inadequate coverage of low-abundance genes. For these overlapping gene sets, global correlation of expression level changes was not observed in bulk mRNA-seq and snRNA-seq data (S3 Fig). However, several genes showed age-dependent gene expression changes in specific cell types, similar to the results in bulk mRNA-seq (Fig 2F, 2G, 2I and 2J). The snRNA-seq samples were obtained from 3- and 24-month-old mice for the hypothalamus, and 4- and 24-month-old mice for the hippocampus. Our bulk RNA-seq samples were obtained from 2- and 15-month-old mice for both

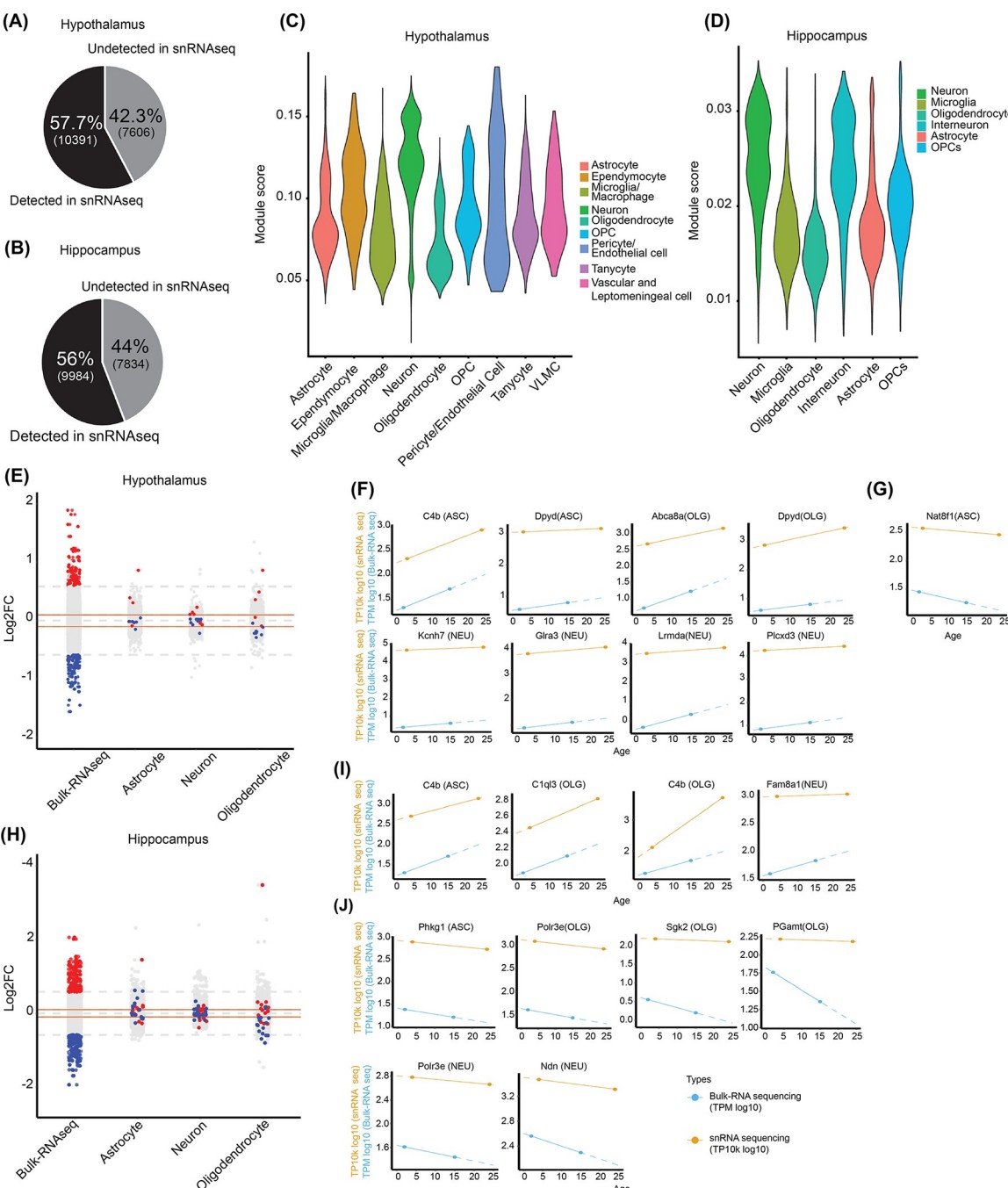

**Fig 2. Comparison of bulk RNA-sequencing and publicly available single-nucleus RNA sequencing datasets.** (A) Pie chart showing total genes expressed in bulk RNA-seq and the proportion of gene coverage compared with snRNA-seq from hypothalamic tissues. (B) Pie chart showing the total number of genes expressed in bulk RNA-seq and the percentage of gene coverage compared with snRNA-seq from hippocampal tissues. (C) Violin plot of bulk RNA-seq illustrating total distribution of covered genes of module scores for astrocytes, ependymocytes, microglia/macrophages, neurons, oligodendrocytes, OPCs, pericytes/endothelial cells, tanycytes, and VLMCs in hypothalamic tissues from snRNA-seq datasets. (D) Violin plot of bulk RNA-seq illustrating total distribution of covered genes of module scores for astrocytes, ependymocytes, microglia/macrophages, neurons, oligodendrocytes, OPCs, pericytes/endothelial cells, tanycytes, and VLMCs in hippocampal tissues from snRNA-seq datasets. Module scores for covered genes in both bulk RNA-seq and snRNA-seq are shown. More scores analyzed from the Seurat R package represent the average expression levels of the genes of interest against that of a set of control genes [57]. (E) Strip chart of DEGs from bulk RNA-seq (cut-off value indicated as gray dashed lines: fold change > 1.5, $p_{adj} < 0.05$) and overlap of these genes with snRNA-seq data (cut-off value indicated as red lines: $\log_2 FC > 0.1$, $p_{adj} < 0.05$) in hypothalamic tissue. Upregulated (red) or downregulated (blue) DEGs from bulk RNA-seq data that are also detected in snRNA-seq data are marked in the chart. (F, G) Expression of individual age-regulated genes normalized to TPM $\log_{10}$ counts from bulk

RNA-seq datasets and normalized TP10k $log_{10}$ counts from snRNA-seq datasets for (F) differentially upregulated and (G) differentially downregulated genes in hypothalamic tissues. (H) Strip chart of DEGs from bulk RNA-seq (cut-off value indicated as gray dashed lines: fold change > 1.5, $p_{adj}$ < 0.05) and overlap of these genes with snRNA-seq data (cut-off value indicated as red lines: $log_2FC$ > 0.1, $p_{adj}$ < 0.05) in hippocampal tissue. Upregulated (red) or downregulated (blue) DEGs from bulk RNA-seq data that were also detected in the snRNA-seq data are marked in the chart. (I, J) Expression of individual age-regulated genes normalized to TPM $log_{10}$ counts from bulk RNA-seq datasets and normalized TP10k $log_{10}$ counts from snRNA-seq datasets for (I) differentially upregulated and (J) differentially downregulated genes in hippocampal tissues.

hypothalamic and hippocampal regions. We plotted the normalized expression levels for each individual gene using the same $log_{10}$-scale as the publicly available snRNA-seq data (TP10k values) and our bulk mRNA-seq data (TPM values) at each aging time point. In hypothalamic tissue, we observed increased mRNA levels of *C4b* and *Dpyd* in astrocytes; *Abca8a* and *Dpyd* in oligodendrocytes; and *Kcnh7*, *Glra3*, *Lrmda*, and *Plcxd3* in neurons in both snRNA-seq and our bulk mRNA-seq data (Fig 2F). *C4b* was previously defined an aging-induced gene and implicated in age-related macular degeneration [11, 59]. The mRNA level of *Nat8f1* in astrocytes decreased during brain aging (Fig 2G). In hippocampal tissue, we observed increased mRNA levels of *C4b* in astrocytes; *C4b* and *C1ql3* in oligodendrocytes; and *Fam8a1*, despite the slight increase, in neurons (Fig 2I). Decreased mRNA levels of *Phkg1* in astrocytes; *Polr3e*, *Sgk2*, and *Gamt* in oligodendrocytes; and *Polr3e* and *Ndn* in neurons were observed during brain aging in the snRNA-seq data (Fig 2J). Combined with the snRNA-seq data of 24-month aged mice, several early signatures of brain aging in bulk mRNA-seq data of 15-month-old aged mice were confirmed in specific cell types of brain tissue.

## Age-dependent microRNA expression changes in hippocampal and hypothalamic tissues

To investigate miRNA-mediated regulatory networks of gene expression during the early stages of brain aging, we performed small-RNA sequencing in the hypothalamus and hippocampal tissues of young (2-month-old) and aged (15-month-old) mice using the same RNA samples as for mRNA sequencing analysis (S4A Fig). A small number of differentially expressed miRNAs in hypothalamic and hippocampal tissues during brain aging were identified (Fig 3A and 3B). Among a total of 731 miRNAs analyzed in hypothalamic tissue, six miRNAs were upregulated, and three miRNAs were downregulated (fold change > 1.5; $p_{adj}$ < 0.05) during brain aging (Fig 3A, S4B, S4F Fig and S2 Table). A total of 731 miRNAs were expressed in hippocampal tissue, of which 16 were upregulated and 29 were downregulated (fold change > 1.5; $p_{adj}$ < 0.05) (Fig 3A, S4C, S4G Fig and S2 Table). Differentially expressed miRNAs miR-542-3p and miR-322-5p were validated by quantitative PCR (S4D and S4E Fig). A previous study, in which 49 upregulated and 31 downregulated miRNAs in hippocampal tissues between young (2 and 6 months) and aged (18 months) mice were detected, reported downregulation of miR-322-5p and miR-322-3p [22]. In this study, three miRNAs, mmu-miR-5100, mmu-miR-8114, and mmu-miR-3084-3p were upregulated, and three miRNAs, mmu-miR-322-3p, mmu-miR-542-3p, and mmu-miR-412-5p were downregulated in both hypothalamic and hippocampal tissues (Fig 3B).

## Differentially expressed miRNAs and their target genes are involved in specific biological pathways during brain aging

To investigate the physiological effects of differentially expressed miRNAs, we first predicted miRNA target genes using Targetscan-Mouse v7.2 [60]. We then selected a subset of target genes that were differentially expressed from our mRNA sequencing data. In hypothalamic

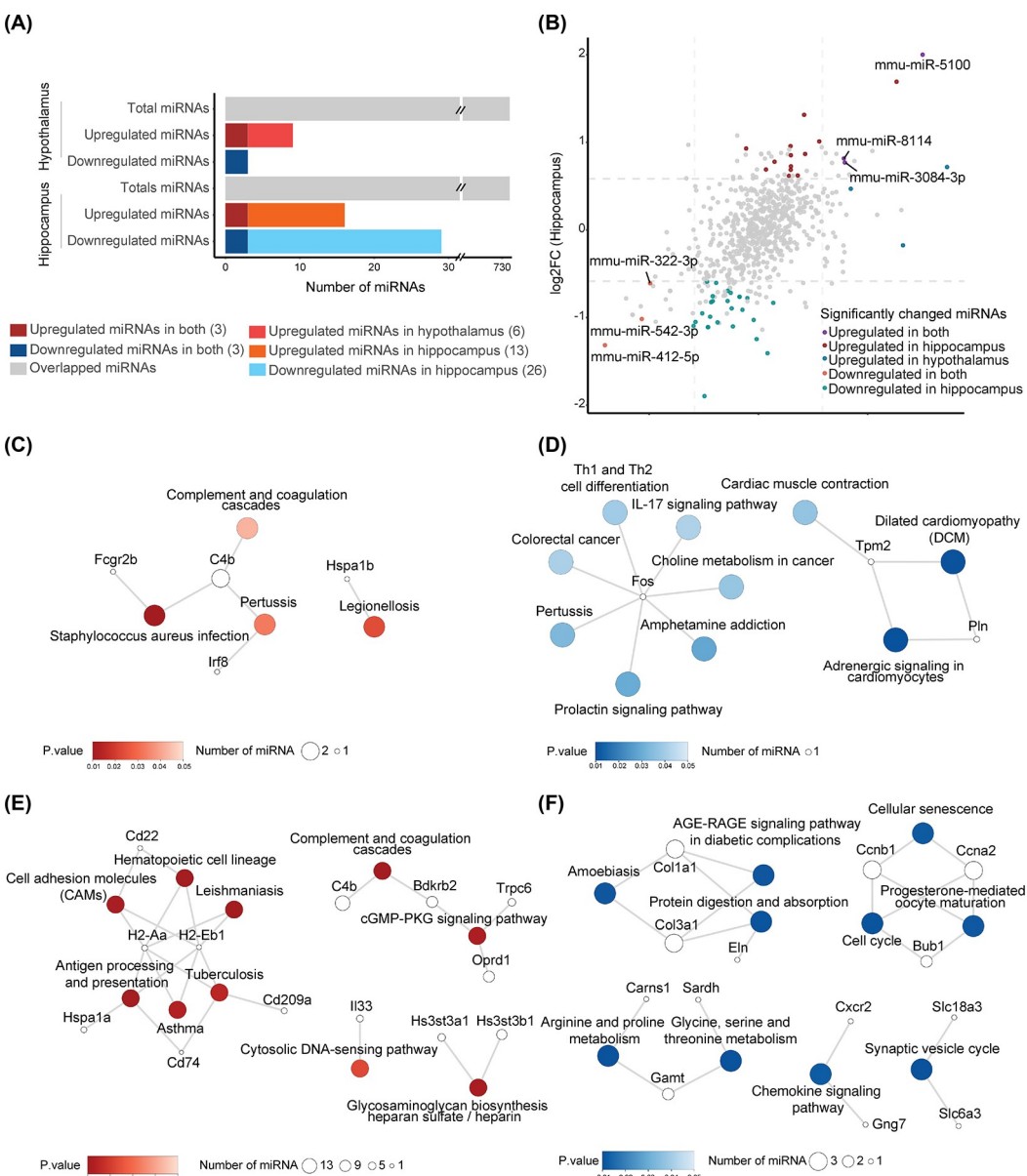

**Fig 3. Differentially expressed miRNAs in the hypothalamus and hippocampus of young and aged mice.** (A) The bar graph shows the number of total detected miRNAs and tissue-specific upregulated and downregulated miRNAs (cut off value: fold change > 1.5, $p_{adj}$ < 0.05) in hypothalamic and hippocampal tissues. All the miRNAs are commonly detected in both tissues. (B) Scatterplot of miRNA expression level changes in hippocampal and hypothalamic tissues during aging. Significantly upregulated and downregulated genes ($p_{adj}$ < 0.05) in the hypothalamic or hippocampal tissues during aging are highlighted with colors. (C–F) KEGG signaling pathways visualized in a network of miRNA-mediated regulatory target genes from mRNA and small-RNA sequencing data. Dot size for each gene indicates the number of differentially expressed miRNAs expected to target DEGs. (C) Downregulated miRNAs expected to target upregulated genes in hypothalamic tissues. (D) Upregulated miRNAs expected to target downregulated genes in hypothalamic tissues. (E) Downregulated miRNAs expected to target upregulated genes in hippocampal tissues. (F) Upregulated miRNAs expected to target downregulated genes in hippocampal tissues.

tissue, 27 upregulated genes were predicted as potential targets for the three downregulated miRNAs, of which *Abca8a*, *C4b* and *Xdh* exhibited increased expression levels of more than two-fold (S5 Fig). For the six upregulated miRNAs, 40 downregulated genes, including *Wdfy1*,

*Aplnr*, *Dynlt1b*, *Nnt*, *Pdlim2*, and *Gabra2* (log2 fold change < -1), were predicted as potential targets (S6 Fig). In hippocampal tissue, 64 upregulated target genes for 26 downregulated miR-NAs (S7 Fig) and 76 downregulated target genes for 15 upregulated miRNAs (S8 Fig) were predicted. Among them, 8 upregulated target genes, *Abca8a*, *C4b*, *Pcdhb9*, *Pcdhb3*, *Xdh*, *Mlph*, *Zbtb16*, and *Zc3hav1*, and 17 downregulated target genes, *Wdfy1*, *Nnt*, *Pdlim2*, *Sgk2*, *Rnf122*, *Gabra2*, *Mog*, *Btbd16*, *Cldn11*, *Mal*, *Eln*, *Tspan2*, *Ddah2*, *Srd5a1*, *Cd248*, *Hist1h1c* and *Apcdd1*, were commonly detected in both hypothalamic and hippocampal tissues. We then analyzed a gene regulatory network of miRNA-targeted DEGs using the Search Tool for the Retrieval of Interacting Genes/Proteins (STRING). We found densely interconnected networks of miRNA target genes, suggesting that miRNAs mediate functional networks through gene expression regulation (S9 Fig). KEGG pathway enrichment analysis of potential miRNA target genes with differential expression during aging also indicated the functional relevance of miRNA-mediated gene expression regulation. In the hypothalamus, upregulated target genes *C4b*, *Fcgr2b*, and *Irf8* are enriched in the complement and coagulation cascades, which are targeted by downregulated miR-322-3p or miR-542-3p in hypothalamic tissues (Fig 3C and S5 Fig). Downregulated genes targeted by upregulated miRNAs were significantly enriched in dilated cardiomyopathy, adrenergic signaling, and prolactin signaling pathways in hypothalamic tissue (Fig 3D). In the hippocampus, upregulated genes targeted by downregulated miRNAs were significantly enriched in the complement and coagulation cascades, as well as antigen processing and presentation signaling pathways (Fig 3E). *Cd22*, which is upregulated in the aged brain, restrains phagocytic capacity and phagocytic clearance of protein aggregates and cellular debris [40]. In our analysis, we found, in addition to *Cd22*, several upregulated genes involved in antigen processing and presentation signaling pathways, which is concordant with the decreased expression of relevant miRNAs (Fig 3E). Downregulated genes targeted by upregulated miRNAs were significantly enriched in the arginine and proline metabolic pathways (Fig 3F). Taken together, our gene network and functional enrichment analyses suggest that miRNA-mediated gene expression regulation induces age-dependent molecular and cellular changes.

## Gene regulatory network of differentially expressed miRNAs and their target genes in aged brain

From our mRNA and small-RNA sequencing data, we identified the network of differentially expressed miRNAs and their predicted target genes that are commonly found in the hypothalamus and hippocampus. Upregulated *Pcdhb3*, *Zc3hav1*, *Pcdhb9*, *C4b*, *Zbtb16*, *Mlph*, *Xdh*, and *Abca8a* are linked to downregulated miRNAs miR-322-3p and miR-542-3p as miRNA–target gene pairs (Fig 4A). Upregulated miRNAs miR-3084-3p and miR-8114, and downregulated target genes *Sgk2*, *Gabra2*, *Nnt*, *Wdfy1*, *Hist1h1c*, *Pdlim2*, *Ddah2*, *Rnf122*, *Cldn11*, and *Tspan2* (Fig 4B) also exhibit miRNA–target gene relationships. We then investigated whether age-related differentially expressed target genes are repressed by their corresponding miRNAs. *C4b* expression levels decreased upon transfection with miR-322-3p and miR-542-3p mimics in the astrocyte cell line (Fig 4C).

Relieving miRNA-mediated suppression of gene expression during brain aging can contribute to *C4b* upregulation, which is a noticeable aging signature in our analysis. *C4b* is a key component of the complement system that plays an important role in innate immunity and links the immune system to the coagulation pathway [62, 63]. Higher *C4b* levels are associated with inflammation and tissue destruction [64, 65]. Dysfunction of the complement and coagulation systems are involved in age-related macular degeneration, thrombosis, and hemorrhage [66]. Our study found aging-dependent changes of miRNAs and target genes involved in the

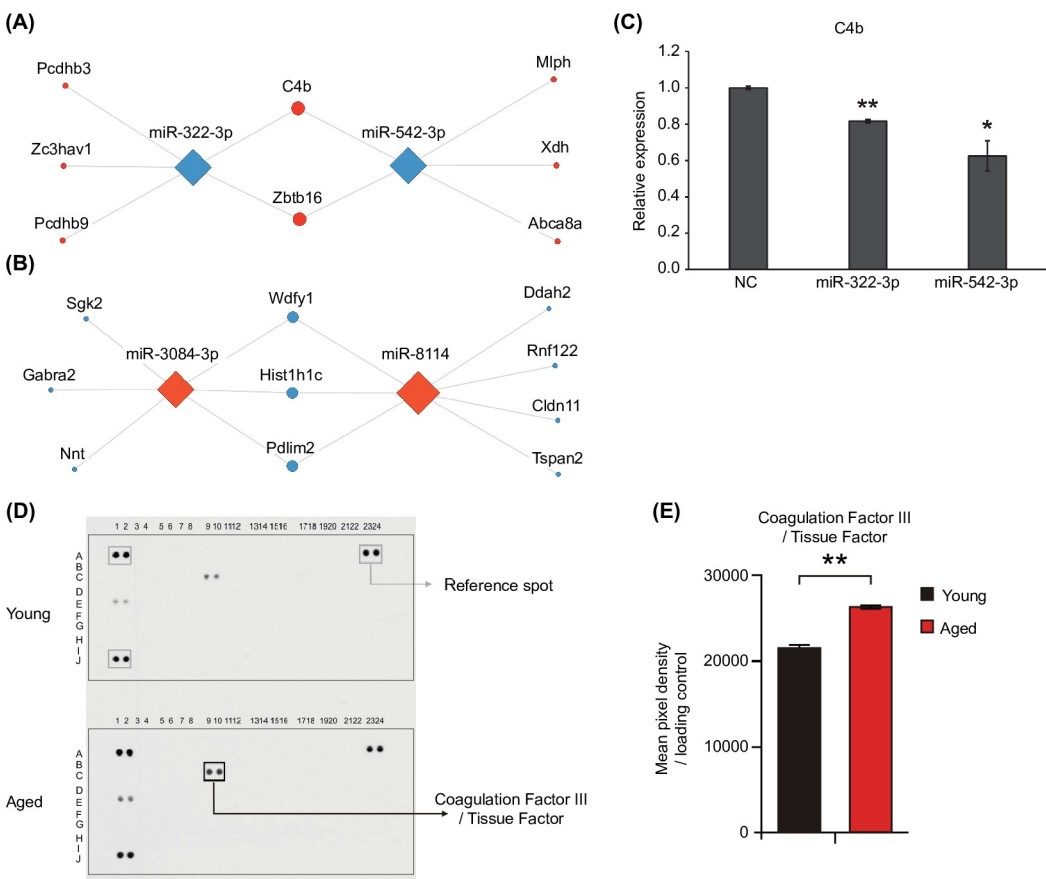

**Fig 4. miRNA-mediated target gene regulation during brain aging.** (A, B) Differentially expressed miRNAs and their predicted common target genes in both hypothalamic and hippocampal tissues. Node sizes for target genes ranked by the degree method [61]. The network of downregulated miRNAs and upregulated target genes is shown in (A). The network of upregulated miRNAs and downregulated target genes is shown in (B). (C) Validation of target gene expression following mimic miR-322-3p and miR-542-3p transfection in the astrocyte cell line by qRT-PCR (n = 3). Data are expressed as mean ± standard deviation (SD). **p < 0.01, * p < 0.05 by Student's two-tailed t-test. (D) Representative images of immunoblot for coagulation factor III and 110 soluble proteins in the brains of young and aged mice. (E) Quantification of coagulation factor III. Data are expressed as mean ± standard deviation (SD). **p < 0.01, * p < 0.05 by Student's two-tailed t-test.

complement and coagulation pathways (Figs 3C, 3E and 4A). Therefore, we examined the alteration of a key effector in the complement and coagulation pathways in aged mouse brain samples. We analyzed the levels of coagulation factor III, a primary initiator of the extrinsic coagulation pathway, and 110 soluble proteins, including cytokines, chemokines, and growth factors, with the Mouse XL Cytokine Array Kit in young and aged brains using the same samples as for mRNA and small-RNA sequencing. The levels of most other cytokines and soluble proteins did not significantly change in the aged brains (Fig 4D). However, higher coagulation factor III levels were observed in the aged brains than in the young brains (Fig 4D and 4E). Collectively, our data suggest dysregulation of complement and coagulation pathways as a prominent feature of gene expression changes during brain aging.

## Discussion and conclusions

miRNA-mediated regulatory networks in the early stages of normal brain aging are not comprehensively understood. We therefore performed simultaneous mRNA and small-RNA

sequencing of hypothalamic and hippocampal tissues from young and aged mouse brains, which enabled identification of novel miRNA-mediated regulatory networks (Fig 1A). The number of genes with significant changes is limited, and the overall ranges of gene expression changes are small in both hypothalamic and hippocampal tissue during the early stage of normal brain aging (2 months and 15 months). Previous studies reported more variability in differentially expressed miRNAs and mRNAs in hippocampal tissue at later stages (2 months and 18 months [22], and 2 months and 24 months [12]) of aging. However, we identified several aging-related differentially expressed mRNAs (S1B and S1C Fig) and miRNAs (S4B, S4C, S4F and S4G Fig) in hypothalamic and hippocampal tissues. The early signatures revealed by our study can provide insight into triggering factors that induce age-dependent progressive molecular and cellular changes, which warrant further investigation. Our RNA-seq data were generated from the hypothalamic and hippocampal tissues of young and aged male mice. Therefore, further analysis of aging in female mouse brain is also needed to identify both common and sex-specific aging processes.

Recently, single-cell sequencing has been performed to understand cell type-specific aging processes in mouse brains [13, 54]. Although snRNA-seq data for the hypothalamus and hippocampus only cover approximately half of the genes analyzed from our bulk mRNA-seq, we tried to estimate the cell-type origin of DEGs in our bulk mRNA-seq data using snRNA-seq data. Only a small number of genes among approximately 100 DEGs of the bulk mRNA-seq data were found in the snRNA-seq data. DEGs that change sensitively in response to cellular signaling or environmental stimulation may be expressed in low abundance and can therefore not be detected by snRNA-seq analysis [67–69]. Several DEGs of our bulk mRNA-seq data similarly showed changed expression in snRNA-seq data, especially in oligodendrocytes, even though the total genes analyzed by bulk mRNA-seq were mainly found in the snRNA-seq data of neurons. Compared with neurons, which comprise approximately 50% of mouse whole brain tissue, oligodendrocytes make up a smaller proportion [70]. However, gene expression changes during aging seem to be particularly dynamic in oligodendrocytes. When expression levels of DEGs detected in our bulk mRNA-seq data were evaluated in snRNA-seq, *C4b* in particular was shown to be upregulated in several cell types, including hippocampal astrocytes and oligodendrocytes, and hypothalamic astrocytes (Fig 2I). Previous studies have reported that isolated hypothalamic [71] and hippocampal astrocytes [24] exhibit increased *C4b* expression during aging. DEGs only detected in bulk mRNA-seq data need to be further validated with a sensitive method to capture small changes at the single cell level, which would further our understanding of cell type-specific changes in gene expression during brain aging.

In this study, we identified specific pathways related to gene expression changes during the early aging process (Fig 1D and 1E). Immune-related pathways were significantly enriched in DEGs. Specifically, our mRNA-seq analysis showed notable upregulation of *Cd22*, previously reported to regulate B cell receptor signaling during aging [40], consistent with another previous report [12]. We also analyzed miRNA-mediated gene regulatory networks using miRNA and mRNA expression profiles in young and aged mouse brains (S9 Fig). miRNA-target genes with differential expression in our sequencing data are enriched in immune-related biological pathways, including the complement and coagulation cascades, the IL-17 signaling pathway, antigen processing and presentation, and chemokine signaling. Giannos and Prokopidis [72] performed a meta-analysis of gene expression data of hippocampal tissues of young (5–8 months) and aged (21–26 months) male rats to identify clustered protein–protein interaction networks of DEGs. Similar to our result, their highly-ranked cluster of 19 age-related hippocampal DEGs was also associated with immune response. These independent analyses of aging in mouse and rats indicate that immune system and immune-related responses are closely

related to the early aging process, which is consistent with previous findings that immune system dysfunction has a great effect on aging [73, 74].

*C4b*, a representative gene in the complement system, was upregulated and targeted by two downregulated miRNAs in the hypothalamus, and thirteen downregulated miRNAs in the hippocampus during brain aging (Fig 3E). In particular, miR-542-3p, miR-322-3p, and *C4b* form a gene regulatory network commonly found in the hypothalamus and hippocampus. Serum concentrations of complement components have previously been reported to increase in the brains of aged mice [75], and are associated with increased susceptibility to neurodegeneration. A knockout mouse model of *C1q*, another component of the complement system [76], exhibited less cognitive decline [77, 78]. The complement system is interconnected with the coagulation system [79], of which coagulation factor III was found to be elevated in our aged mouse brain sample at the protein level. Aging is associated with thrombosis and bleeding disorders [64–66]. In addition, thrombin dysregulation leads to inflammatory brain diseases such as Alzheimer's disease [80–82]. Zorzetto et al. [83] found that patients with Alzheimer's disease have statistically significant increased C4b expression compared with healthy controls, and suggested that increased C4b explain the possible role of elevated complement component C4 protein levels in Alzheimer's disease development. Stein et al. [84] indicated that elevated C4b level may be associated with increased inflammation in aging brains. Taken together with previous findings, our data indicate the necessity of clearly defining the direct effects of a gene regulatory network on the complement and coagulation cascades, which could uncover the critical gene regulatory network of miRNAs and their target genes in brain aging.

Herein, we presented mRNA and miRNA profiles generated from the same samples of mouse hypothalamus and hippocampus tissues, which covered age-dependent changes from 2 to 15 months. Our data provides a valuable resource to understand miRNA-mediated gene expression regulation during the early stage of brain aging. In addition, the identification of other potential age-dependent regulators of gene expression, such as circular RNA and long non-coding RNA, could reveal complex gene regulatory networks involved in aging processes.

## Materials and methods

### Animal studies

All experimental protocols were approved by the Institutional Animal Care and Use Committee of Soonchunhyang University (protocol number #SCH19-0027). Male C57BL/6J mice (2 months old for young, n = 6; 15 months old for aged, n = 6) were obtained from ORIENT BIO (Seongnam-si, Korea) and Korea Basic Science Institute (KBSI) Gwangju Center (Gwangju, Korea). Each cage contained 5–6 mice. Mice were fed Harlan 2018S (ENVIGO Co., LTD) ad libitum. The feed was sterilized using irradiation and hydrogen peroxide ($H_2O_2$). Water and Beta-Chip bedding (Abedd, LTE E-001) were autoclaved at 121°C for 20 minutes before use. Mice were housed in specific-pathogen free environment. Temperature and humidity were maintained at 18–22°C and 40–60%, respectively. Ventilation rate was 10–15 times/hour (All Fresh Air System). Mice were kept in 12 h light (7:00–19:00)/12 h dark cycle (19:00–7:00). We sterilized the equipment used using an autoclave, and goods that were non-autoclavable were sterilized with $H_2O_2$.

### Brain tissue isolation

We isolated the hypothalamus and hippocampus of the mouse brains. Tissues were dissociated with Tissue Extraction Reagent I (FNN0071, Thermo Fisher Scientific, Waltham, MA, USA) or TRIzol™ Reagent (15596026, Thermo Fisher Scientific) for 1 min using a BeadBug™ microtube homogenizer (Z763713, Sigma-Aldrich, St Louis, MO, USA). Total RNA was isolated from hypothalamus and hippocampus tissue lysates using TRIzol™ Reagent (15596026, Thermo Fisher Scientific).

## Cell lines and transfection

Astrocyte cell line C8-D1A (astrocyte type I clone from C57/BL6 strains) was purchased from the American Type Culture Collection (ATCC, Manassas, VA, USA). Cells were cultured in Dulbecco's modified Eagle's medium (DMEM, Thermo Fisher Scientific) supplemented with 10% fetal bovine serum (FBS, Thermo Fisher Scientific) and incubated at 37˚C and 5% $CO_2$. Cells were subsequently transfected with 40 nM mmu-miR-542-3p, mmu-miR-322-3p, and NC mimics (GenePharma, Shanghai, China) using Lipofectamine (RNAiMAX Invitrogen, Carlsbad, CA, USA) according to the manufacturer's instructions. Cells were harvested after 48 hours of transfection for the analysis.

## Quantitative RT-PCR

Total RNA was extracted from cells using TRIzol™ Reagent (15596026, Thermo Fisher Scientific) and DNase I. cDNA synthesis was performed with 0.5 µg total RNA using ReverTra Ace qPCR RT Master Mix reverse transcription kit (Toyobo, Osaka, Japan). Using the synthesized cDNA, quantitative PCR (qPCR) was used to determine mRNA levels, normalized to universal housekeeping reference gene *36B4* levels, and qPCR was performed using SYBR Green Real-time PCR Master Mix (Toyobo). Mature miRNA levels were identified using TaqMan micro-RNA assays (Applied Biosystems, Bedford, MA, USA) and normalized to *U6* snRNA levels. The comparative cycle threshold (Ct) method was used to determine relative mature miRNA and mRNA levels. Primer sequences used for RT-PCR were as follows: 36B4, forward 5′– AACGGCAGCATTTATAACCC–3′, reverse 5′–CGATCTGCAGACACACACTG–3′; C4b, forward 5′–ACTTCAGCAGCTTAGTCAGGG–3′, reverse 5′–GTCCTTTGTTTCAGGGGACAG–3′.

## Analysis of coagulation factor III levels

Whole male mouse brains were isolated and lysed in Tissue Extraction Reagent I (FNN0071, Thermo Fisher Scientific) to quantify coagulation factor III levels. Tissue lysates were centrifuged at $15,300 \times g$ for 10 min at 4˚C, and the supernatant protein concentrations determined using the Bradford assay (500–0006, Bio-Rad Laboratories, Hercules, CA, USA). Mouse brain protein lysates were analyzed for coagulation factor III and 110 soluble proteins, including chemokines, cytokines, and growth factors, using the Mouse XL Cytokine Array Kit (ARY028, R&D systems, Minneapolis, MN, USA), according to the manufacturer's instructions. Briefly, 100 µg protein lysate was incubated with nitrocellulose membranes containing 111 different captured antibodies, printed in duplicate, for 16 h at 4˚C. Nitrocellulose membranes were subsequently incubated with a detection antibody diluted in assay buffer for 2 h at room temperature and then incubated with streptavidin–horseradish peroxidase (HRP) in assay buffer for 30 min at room temperature. Immunoreactive spots on the nitrocellulose membranes were detected using the chemical reagent mix and then exposed to X-ray film. In this study, multiple exposure times were used. Pixel densities from positive signals on the developed X-ray film were collected using a transmission mode scanner and analyzed with image analysis software (HLImage++ Version 25.0.0r, https://www.wvision.com/QuickSpots.html, Western Vision Software, Salt Lake City, UT, USA). Pixel densities were quantified and the relative change in analyte levels was determined.

## mRNA library preparation and sequencing

Three biological replicates (three mice) were prepared from each group of young (2-month-old) and aged (15-month-old) male mice. Total RNA was extracted from the mouse brain tissue using TRIzol (Invitrogen) reagent. For bulk-RNA sequencing, a 2100 Bio-Analyzer and

RNA 6000 Nano Kit (Agilent, Santa Clara, CA, USA) were used to assess RNA quantity and quality. Total RNA from each sample was used to generate mRNA sequencing libraries provided by BGI Genomics Co., Ltd. Briefly, poly-A-containing mRNA molecules were purified using poly-T oligo-attached magnetic beads. After purification, the mRNA was cleaved with divalent cations at high temperature, and the cleaved RNA fragments were subsequently converted into single-strand cDNA using reverse transcriptase and random primers. The second strand of cDNA was synthesized using DNA Polymerase I and RNase H. After PCR amplification, the products were purified and enriched. DNA nanoballs were loaded into patterned nanoarrays, and 100 bp pair-end reads were performed using the BGISEQ-500 platform for further data analysis.

### Small-RNA library preparation and sequencing

Small-RNA sequence libraries were prepared using the TruSeq Small RNA Library Prep Kit (Illumina, San Diego, CA, USA), with 5 μg total RNA per sample for 12 samples. Sequencing adapters were attached to the RNA molecules, which were used as primer-binding sites during reverse transcription and PCR amplification of the cDNA sequence pool. The amplified cDNA fragments were separated on agarose gel, and the band corresponding to the miRNA fragments with ligated adapters was excised for subsequent sequencing. The library ranged from 145 to 160 bp. cDNA fragments were sequenced by read-length using the sequence-by-synthesis method on an Illumina HiSeq 2500 platform.

### mRNA sequence analysis

For mRNA sequencing, low-quality raw sequence reads, reads with adapters, and reads with unknown bases were filtered using SOAPnuke software (https://github.com/BGI-flexlab/SOAPnuke) to yield clean reads [85]. Next, filtered clean reads were mapped onto the reference mouse genome (*Mus musculus*, UCSC mm10) (https://genome.ucsc.edu/) using Hierarchical Indexing for Spliced Alignment of Transcripts (HISAT) (http://www.ccb.jhu.edu/software/hisat/index.shtml) [86]. StringTie (http://ccb.jhu.edu/software/stringtie) was used to predict novel genes [87]. Cuffcompare (Cufflinks tools) (http://cole-trapnell-lab.github.io/cufflinks) [88] was used to compare the reconstructed transcripts to the reference annotation, and CPC tools (http://cpc.cbi.pku.edu.cn) [89] were used to predict the coding potential of novel transcripts. Subsequently, the gene expression levels of each sample were determined using Bowtie2 (http://bowtie-bio.sourceforge.net/Bowtie2/index.shtml) [90] and RSEM (http://deweylab.biostat.wisc.edu/RSEM) [91]. To reflect gene expression correlations between samples, we calculated the Pearson correlation coefficients of all gene expression levels and displayed these coefficients using heat maps. All samples were hierarchically clustered according to the expression levels of all genes.

### Small-RNA sequence analysis

Raw sequencing reads were filtered based on quality and trimmed using adapter sequences. The trimmed reads form a unique cluster that matches both sequence identity and read length. The reads were aligned to the reference mouse genome (*Mus musculus*, UCSC mm10) (https://genome.ucsc.edu/) and miRBase v21 (http://www.mirbase.org/) [92] precursor miRNAs to identify known and novel miRNAs, respectively. The miRDeep2 (https://www.mdc-berlin.de/8551903/en/) [93] (Friedländer, et al., 2008) algorithm was used to predict potential hairpin structures. Unique cluster reads were then sequentially matched against the reference mouse genome, miRBase v21, and the non-coding RNA database, Rfam9.1 (http://rfam.xfam.org/) [94], to classify the known miRNAs. Read counts for each miRNA were extracted from the

mapped miRNAs to determine abundance. Reproducibility was evaluated using Pearson's coefficient of the $\log_2$(Count+1)-value. For the range $-1 \leq r \leq 1$, values approaching 1 represent greater similarity between samples. Differentially expressed miRNAs in young and aged mice were identified using DESeq2 algorithms (https://github.com/mikelove/DESeq2) [95].

### Differential gene expression analysis

For small-RNA sequencing, the reads for each miRNA were normalized to the total reads of each sample. The normalized values were then converted to reads per million (RPM), calculated as miRNA count/total count of each sample x 1 million, to identify differentially expressed miRNAs. Fold-change and nbinomWald Test from DESeq2 algorithms were used for statistical analysis of each comparison pair. Selection criteria for differentially expressed mRNA and miRNAs included fold change $\geq 1.5$ and an adjusted p-value $< 0.05$. Volcano plots for mRNA sequencing were generated using the EnhancedVolcano v1.12 R package (https://github.com/kevinblighe/EnhancedVolcano) [96]. Scatter plots for small-RNA sequencing were generated using the ggplot2 R package (https://ggplot2.tidyverse.org).

### Comparison of individual age-related genes in bulk RNA-seq and snRNA-seq datasets

We compared changes in the expression of individual candidate genes during brain aging in hypothalamic and hippocampal tissues from publicly available snRNA-seq hypothalamic (GSE188646) (https://www.ncbi.nlm.nih.gov/geo/query/acc.cgi?acc=GSE188646) and hippocampal (GSE161340) (https://www.ncbi.nlm.nih.gov/geo/query/acc.cgi?acc=GSE161340) datasets to our bulk RNA-seq datasets. RNA counts from each cell type were normalized and transcripts per 10,000 (TP10k)-values were calculated using the Seurat R package (https://satijalab.org/seurat/) [97]. We then used the $\log_{10}$ scale to normalize the TP10K values of each transcript of the identified cell types in snRNA-seq datasets. For our bulk RNA-seq datasets, which were generated from hypothalamic and hippocampal tissues, we used the fragments per kilobase of transcript per million fragments mapped (FPMK)-count of each identified gene and divided it by the gene length. We then multiplied these values by one million and divided by the total number of identified genes to calculate normalized transcript per million (TPM) values. We then used the $\log_{10}$ scale-normalized TPM values from each transcript to compare individual RNA counts in the publicly available snRNA-seq datasets and our bulk RNA-seq datasets. To analyze common genes in both bulk mRNA-seq and snRNA-seq data within the identified cell types, we used module score function (Seurat::AddModuleScore()) from the Seurat R package, which represents average expression level of each cluster on single-cell level, subtracted by the aggregated expression of control sets [57].

### miRNA target prediction and functional analysis

Targetscan-Mouse v7.2 (https://www.targetscan.org/mmu_72/) [60] was used for target prediction analysis of the differentially expressed miRNAs. Predicted target genes from differentially expressed miRNAs were analyzed with KEGG pathway enrichment (https://www.genome.jp/kegg/kegg1b.html) [98] using the "pathfindR" R package (https://github.com/egeulgen/pathfindR) [99]. Statistical significance was set at $p < 0.05$. Enriched KEGG pathways and differentially up- and downregulated target genes were constructed and visualized using Cytoscape v3.9.0-BETA1 software (https://cytoscape-builds.ucsd.edu/Cytoscape-3.9.0/beta1/) [100]. Differentially expressed miRNAs and mRNA in hippocampal and hypothalamic tissues were also visualized using the Cytoscape software.

## miRNA–mRNA regulatory network and core subnetwork analysis

Upregulated and downregulated differentially expressed target genes were mapped into the STRING (Search Tool for the Retrieval of Interacting Genes/Proteins) (https://string-db.org/) database (stringApp v1.7.1) (https://github.com/RBVI/stringApp/releases) [101], and a confidence score $\geq 0.1$ was used to assess the information of the gene regulatory network between target genes visualized with Cytoscape v3.9.0-BETA1. The Cytoscape plug-in app MCODE v2.0 (https://mcode.readthedocs.io/en/latest/) [102] was applied to the gene regulatory networks to calculate the major hub target genes, and the top ten targets were ranked by the degree method using the Cytoscape plugin app cytoHubba v0.1 (https://apps.cytoscape.org/apps/cytohubba) [61] to identify significant functional modules.

## Supporting information

**S1 Fig. mRNA sequencing analysis of the hypothalamus and hippocampus of young and aged mice.** (A). Heatmap of Pearson's correlation coefficients for gene expression in each young and aged mouse brain hypothalamus and hippocampus sample. The X and Y axes represent each sample, and the color represents the correlation coefficient. (B, C). Volcano plots of all detected genes from mRNA sequencing analysis of young versus aged hypothalamic and hippocampal tissues. Differentially upregulated genes are colored red, differentially downregulated genes are colored blue, and non-significant genes are colored gray. The cut-off was set as fold change > 1.5, and statistical significance was set as $p_{adj} < 0.05$.
(TIF)

**S2 Fig. Number of gene distributions covered for cell types from hypothalamic and hippocampal tissue.** (A) Bar graph showing the number of gene distributions covered for each cell type in hypothalamic tissue in bulk RNA sequencing versus single nucleus RNA sequencing datasets. (B) Bar graph showing the number of gene distributions covered for each cell type in hippocampal tissue in bulk RNA sequencing versus single nucleus RNA sequencing datasets.
(TIF)

**S3 Fig. Correlation analysis of differentially expressed genes in both our bulk mRNA sequencing and publicly available single nucleus RNA sequencing datasets.** (A). Pearson's correlation analysis of $Log_2FC$ expression of differentially upregulated genes in bulk mRNA sequencing (cut-off value: fold change > 1.5, $p_{adj} < 0.05$) and single nucleus RNA sequencing (cut-off value: $log_2FC > 0.1$, $p_{adj} < 0.05$) datasets from hypothalamic tissue. Linear regression is indicated with the colored line. The Pearson's correlation coefficient ($R^2$) and p-value are displayed in the graph. Gene expression changes in hypothalamic astrocytes ($R^2 = 0.65$, p = 0.098), oligodendrocytes ($R^2 = 0.01$, p = 0.85) and neurons ($R^2 = 0.01$, p = 0.78) are not correlated. (B) Pearson's correlation analysis of $Log_2FC$ expression of differentially downregulated genes in bulk mRNA sequencing (cut-off value: fold change > 1.5, $p_{adj} < 0.05$) and single nucleus RNA sequencing (cut-off value: $log_2FC > 0.1$, $p_{adj} < 0.05$) datasets from hypothalamic tissue. Linear regression is indicated with the colored line. The Pearson's correlation coefficient ($R^2$) and p-value are displayed in the graph. Gene expression changes in hypothalamic astrocytes ($R^2 = 0.01$, p = 0.81), oligodendrocytes ($R^2 = 0.04$, p = 0.61) and neurons ($R^2 = 0.15$, p = 0.3) are not correlated. (C) Pearson's correlation analysis of $Log_2FC$ expression of differentially upregulated genes in bulk mRNA sequencing (cut-off value: fold change > 1.5, $p_{adj} < 0.05$) and single nucleus RNA sequencing (cut-off value: $log_2FC > 0.1$, $p_{adj} < 0.05$) datasets from hippocampal tissue. Linear regression is indicated with the colored line. The Pearson's correlation coefficient ($R^2$) and p-value are displayed in the graph. Gene expression changes in hippocampal astrocytes ($R^2 = 0.84$, p = 2.9e-05) and oligodendrocytes ($R^2 = 0.34$, p = 0.029)

are significantly correlated, whereas gene expression changes in hippocampal neurons ($R^2 =$ 0.0044, p = 0.81) are not correlated. (D) Pearson's correlation analysis of $Log_2FC$ expression of differentially downregulated genes in bulk mRNA sequencing (cut-off value: fold change > 1.5, $p_{adj} < 0.05$) and single nucleus RNA sequencing datasets (cut-off value: $log_2FC > 0.1$, $p_{adj} < 0.05$) from hippocampal tissue. Linear regression is indicated with the colored line. The Pearson's correlation coefficient ($R^2$) and p-value are displayed in the graph. Gene expression changes in hippocampal astrocytes ($R^2 = 0.00035$, p = 0.95), oligodendrocytes ($R^2 = 0.003$, p = 0.84), and neurons ($R^2 = 0.0061$, p = 0.76) are not correlated.
(TIF)

**S4 Fig. miRNA sequencing analysis of the hypothalamus and hippocampus of young and aged mice.** (A) Heatmaps of Pearson's correlation coefficients of $log_2(Count+1)$ values for miRNA expression of each young and aged mouse brain from hypothalamic and hippocampal tissue samples. The X and Y axes represent each sample, and the color represents the correlation coefficient. (B, C) Volcano plots of detected miRNA expression from miRNA sequencing datasets of young and aged mouse brain from (B) hypothalamic and (C) hippocampal tissues. Differentially expressed genes (cutoff value: fold change > 1.5, $p_{adj} < 0.05$) are colored red for upregulated genes, blue for downregulated genes, and gray for non-significant genes. (D, E) Validation of miRNA expression from small-RNA sequencing datasets by qRT-PCR in the (D) hypothalamic and (E) hippocampal tissues (n = 4). Data are presented as mean ± SEM. **p < 0.01, *p < 0.05. (F, G) Heatmap of differentially expressed miRNAs (cut-off value: fold change > 1.5, $p_{adj} < 0.05$) count numbers (TPM $log_2$) in young and aged mouse brain samples from (F) hypothalamic and (G) hippocampal tissues.
(TIF)

**S5 Fig. Differentially downregulated miRNAs predicted to target differentially upregulated mRNAs in hypothalamic tissues during aging.** Heatmap shows differentially upregulated gene expression (cut-off value: fold change > 1.5, $p_{adj} < 0.05$) predicted to be targeted by differentially downregulated miRNAs (cut off value: fold change > 1.5, $p_{adj} < 0.05$) in mRNA sequencing and small-RNA sequencing datasets from hypothalamic tissue.
(TIF)

**S6 Fig. Differentially upregulated miRNAs predicted to target differentially downregulated mRNAs in hypothalamic tissues during aging.** Heatmap shows differentially downregulated gene expression (cut-off value: fold change > -1.5, $p_{adj} < 0.05$) predicted to be targeted by differentially upregulated miRNAs (cut-off value: fold change > 1.5, $p_{adj} < 0.05$) in mRNA sequencing and small-RNA sequencing datasets from hypothalamic tissue.
(TIF)

**S7 Fig. Differentially downregulated miRNAs predicted to target differentially upregulated mRNAs in hippocampal tissues during aging.** Heatmap shows differentially upregulated gene expression (cut-off value: fold change > 1.5, $p_{adj} < 0.05$) predicted to be targeted by differentially downregulated miRNAs (cut-off value: fold change > 1.5, $p_{adj} < 0.05$) in mRNA sequencing and small-RNA sequencing datasets from hippocampal tissue.
(TIF)

**S8 Fig. Differentially upregulated miRNAs predicted to target differentially downregulated mRNAs in hippocampal tissues during aging.** Heatmap shows differentially downregulated gene expression (cut-off value: fold change > -1.5, $p_{adj} < 0.05$) predicted to be targeted by differentially upregulated miRNAs (cut-off value: fold change > 1.5, $p_{adj} < 0.05$) in mRNA

sequencing and small-RNA sequencing datasets from hippocampal tissue.
(TIF)

**S9 Fig. Protein–protein interaction (PPI) network of miRNA-mediated differentially expressed target genes in hypothalamic and hippocampal tissues during aging.** (A) PPI analysis of differentially upregulated genes targeted by differentially downregulated miRNAs in hypothalamic tissues, derived from mRNA sequencing datasets and small-RNA sequencing datasets. PPI network analyzed by STRING and MCODE, node sizes for target genes are determined by the degree method, and node color indicates the cell type. (B) PPI analysis of differentially downregulated genes targeted by differentially upregulated miRNAs in hypothalamic tissues, derived from mRNA sequencing datasets and small-RNA sequencing datasets. PPI network analyzed by STRING and MCODE, node sizes for target genes are determined by the degree method, and node color indicates the cell type. (C) PPI analysis of differentially upregulated genes targeted by differentially downregulated miRNAs in hippocampal tissues, derived from mRNA sequencing datasets and small-RNA sequencing datasets. PPI network analyzed by STRING and MCODE, node sizes for target genes are determined by the degree method, and node color indicates the cell type. (D) PPI analysis of differentially downregulated genes targeted by differentially upregulated miRNAs in hippocampal tissues, derived from mRNA sequencing datasets and small-RNA sequencing datasets. PPI network analyzed by STRING and MCODE, node sizes for target genes are determined by the degree method, and node color indicates the cell type.
(TIF)

**S1 Table. mRNA sequencing datasets from hippocampal and hypothalamic tissues of young and aged mouse brain during aging.**
(XLSX)

**S2 Table. miRNA sequencing datasets from hippocampal and hypothalamic tissues of young and aged mouse brain during aging.**
(XLSX)

**S1 Dataset. Minimal data set for Fig 4D.**
(TIF)

**S2 Dataset. Minimal data sets for Fig 4C, 4E, S4D and S4E Fig.**
(XLSX)

## Acknowledgments

We thank the members of our laboratories for discussions and assistance throughout this project.

## Author Contributions

**Conceptualization:** Choijamts Munkhzul, Sun Shin Yi, Jong-Seok Moon, Mihye Lee.

**Formal analysis:** Choijamts Munkhzul, Sun Shin Yi, Junhyung Kim, Seongsoo Lee, Jong-Seok Moon, Mihye Lee.

**Investigation:** Choijamts Munkhzul, Sun Shin Yi, Junhyung Kim, Seongsoo Lee, Hyuntae Kim, Jong-Seok Moon, Mihye Lee.

**Resources:** Hyuntae Kim.

**Supervision:** Jong-Seok Moon, Mihye Lee.

**Writing – original draft:** Choijamts Munkhzul, Jong-Seok Moon, Mihye Lee.

**Writing – review & editing:** Junhyung Kim, Seongsoo Lee, Hyuntae Kim.

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
