## [Decision Letter · Decision Letter 0]

28 Jun 2023

PONE-D-23-13060The microRNA-mediated gene regulatory network in the hippocampus and hypothalamus of the aging mousePLOS ONE

Dear Dr. Lee,

Thank you for submitting your manuscript to PLOS ONE. After careful consideration, we feel that it has merit but does not fully meet PLOS ONE’s publication criteria as it currently stands. Therefore, we invite you to submit a revised version of the manuscript that addresses the points raised during the review process.

Based on the reviewers' suggestions, the paper needs major revision.  The reviewers' comments can be found below.

We look forward to receiving your revised manuscript.

Kind regards,

Tanja Grubić Kezele, Ph.D., M.D.

Academic Editor

PLOS ONE

“This work was supported by the National Research Foundation of Korea (Grant NRF-

2021R1A2C4002421 to M.L.; 2019M3E5D3073092 to M.L.; 2021R1C1C1007810 to J.M.) and

Soonchunhyang University Research Fund (Soonchunhyang University Research Fund 2021 to J.M.).”

“We thank the members of our laboratories for discussions and assistance throughout this

project. This work was supported by the National Research Foundation of Korea (Grant NRF-

2021R1A2C4002421 to M.L.; 2019M3E5D3073092 to M.L.; 2021R1C1C1007810 to J.M.) and

Soonchunhyang University Research Fund (Soonchunhyang University Research Fund 2021 to J.M.).”

“This work was supported by the National Research Foundation of Korea (Grant NRF-

2021R1A2C4002421 to M.L.; 2019M3E5D3073092 to M.L.; 2021R1C1C1007810 to J.M.) and

Soonchunhyang University Research Fund (Soonchunhyang University Research Fund 2021 to J.M.).”

Reviewers' comments:

Reviewer's Responses to Questions

**Comments to the Author**

1. Is the manuscript technically sound, and do the data support the conclusions?

Reviewer #1: Yes

Reviewer #2: Yes

2. Has the statistical analysis been performed appropriately and rigorously? 

Reviewer #1: Yes

Reviewer #2: Yes

3. Have the authors made all data underlying the findings in their manuscript fully available?

Reviewer #1: Yes

Reviewer #2: Yes

4. Is the manuscript presented in an intelligible fashion and written in standard English?

Reviewer #1: Yes

Reviewer #2: Yes

5. Review Comments to the Author

Reviewer #1: In this study, the authors aimed to elucidate molecular mechanisms involved in mouse brain aging through the profiling of mRNA and miRNA changes in the hypothalamus and hippocampus of young and aged mice. Moreover, utilizing published scRNA-seq data from the regions under study, they assigned a subset of the differentially expressed genes to different brain cell types and highlighted increased levels of C4b in astrocytes in both brain regions, concomitantly providing independent validation of their findings through the scRNA-seq data used. Then, they identified common miRNA changes in the hypothalamus and hippocampus and experimentally validated reduced levels of miR-542-3p and miR-322-5p in both regions. They proceeded with predicting the targets of differentially expressed miRNAs in conjunction with their transcriptomic data, focusing on transcripts displaying inverse expression patterns to their predicted regulatory miRNAs. Next, they identified the interconnections of these miRNA-regulated transcripts, highlighting a common hypothalamic and hippocampal signature entailing the involvement of the complement and coagulation cascades, in which C4b is involved. They provide functional evidence on the regulatory role of miR-322-3p and miR-542-3p on C4b expression levels, through in vitro analyses and show that coagulation factor III, involved in the coagulation pathway, is increased in aged mice.

This study highlights a functional miRNA-miRNA network, possibly mediated by astrocytes, which affects the coagulation pathway during aging. To this end, this study further extends previous research reporting C4b increases during aging by providing a miRNA-mediated mechanistic insight.

Below are some comments that could help improve the manuscript:

- The text, especially the introduction part, could be more concise to help the reader. I would suggest the introduction to be more focused. Avoid citing several, in most cases very old references, that obscure the reader. Please carefully, revise the references, as in several cases they are not cited in the most appropriate position in the manuscript, and reduce the references cited by keeping the most relevant ones.

- As the authors have stated, their RNA-seq and small RNA-seq analysis identified a relatively low number of differentially expressed genes and miRNAs compared to similar studies. It would be nice if they provide an explanation of possible reasons, that could be for example related with the differential expression analysis packages they used. DeSeq2, used for the miRNA analysis in this study, is commonly used for differential expression analysis. In the results section, instead of commenting on selected transcripts (Cd22, Irak3), it would be better to compare their findings with previous similar studies (for example ref 10 and https://doi.org/10.1016/j.gpb.2020.12.001 for mRNA analyses) and provide a schematic representation (e.g. Venn diagrams) of the detected overlaps.

- The authors highlighted C4b upregulation during aging, which has also been reported in neurodegenerative and neurological disorders, such as Alzheimer’s and Schizophrenia. To my opinion it would be interesting to include in the discussion a kind of parallelism of this finding between aging and neurodegenerative / neurological disorders.

- Regarding materials and methods: Please provide details on the miRNA mimics used for the transfections and state at which point following transfection the cells were harvested for further analysis. Please provide the primer sequences used for C4b qRT PCR and comment on the selection of the housekeeping gene used. Also, provide a description either in the materials and methods or in the corresponding figure legend (Figure 2) of the module score.

Regarding figures: Figure 3: E the statistics symbol is missing. If there is no statistical significance in the result please note it in the legend. C: include the statistic symbols (* , **) meaning to the legend. Figure S4: D: missing the graph for miR-322-5p validation E: please check spelling. The figure shows miR-522-3p. Also, the statistics symbols are missing from E. If no statistical significance, please note it in the legend. Include the statistic symbols (**) meaning to the legend

Reviewer #2: The manuscript provides a comprehensive and meticulous analysis of age-related changes in gene expression and microRNA (miRNA) profiles in the hypothalamus and hippocampus regions of the mouse brain. The authors employed a robust experimental design using both young and aged mouse groups, coupled with both mRNA and small-RNA sequencing. The use of well-established bioinformatics tools like HISAT, StringTie, and Bowtie2 ensured accurate mapping, prediction, and comparison of novel genes and transcripts. Their rigorous approach to data normalization and determination of differentially expressed genes is commendable.

Moreover, the analysis of both mRNA and miRNA provides a broader view of the changes occurring at the genetic level, allowing for a more comprehensive understanding of the aging process. The researchers also explored the miRNA-mRNA regulatory network, a crucial factor in understanding the complex interplay of gene expression regulation. Their use of predictive analysis tools, like Targetscan-Mouse v7.2 and the STRING database, lends weight to their findings and provides potential avenues for further research.

However, the study has some limitations. First, the study used only male mice, limiting the generalizability of the results to female mice or other species. Sex-specific differences in gene expression could impact the aging process, so a more comprehensive study including both sexes might provide a broader perspective. Secondly, while the authors have measured a substantial number of variables, it might have been beneficial to have more biological replicates in each group to increase statistical power and the reliability of the findings.

Furthermore, while the authors have used in vitro studies using astrocyte cell lines, integrating data from a wider range of neuronal and non-neuronal cell types could be beneficial, given the cellular diversity of the brain and the potential influence of other cell types on aging processes.

Overall, while the manuscript is robust in its analysis and provides valuable insights into the aging process at the molecular level, further research, including both sexes and a wider array of cell types, could be beneficial to present a more comprehensive picture of age-related changes in the brain.

Moreover, the authors' representation of the current literature appears incomplete and could be greatly enriched by referencing recent relevant work in this field, particularly studies focusing on the aging brain in other species. A recent gene-expression meta-analysis (doi: 10.3389/fnins.2022.915907) explored the changes from multiple gene expression studies of the hippocampus in aging rats from a molecular interaction perspective and found substantial alterations related to immune functions and immunoglobulin dynamics. This lends credence to the concept of inter-species commonalities in brain aging processes and should be mentioned in the Introduction and / or the Discussion. It is essential for the authors to contextualize their study within the broader framework of aging research.

From a proofreading perspective, the manuscript is generally well-written and organized. The language is clear, concise, and uses standard scientific terminology correctly. The methods section follows a logical progression, detailing the steps of the analysis sequentially and making it easier to follow.

However, there are a few areas that could be improved for clarity and precision. The authors should consider using a more consistent style when referring to various software packages, databases, and resources. For instance, in some cases, URLs are provided, while in others, they are not. This could potentially cause confusion for readers trying to replicate the study. Also, the manuscript often presents several methods sequentially without much transition or explanation of why each step is necessary. Providing more context or rationale for each step of the method might improve clarity for readers unfamiliar with the specific techniques used. Lastly, sentences are quite long and packed with information. Breaking these up into smaller, more digestible sentences might enhance readability. For instance, the sentence starting with "For small-RNA sequencing..." could be split into two for clarity.

6. PLOS authors have the option to publish the peer review history of their article (what does this mean?). If published, this will include your full peer review and any attached files.

Reviewer #1: No

Reviewer #2: No

---

## [Author Response · Author response to Decision Letter 0]

14 Aug 2023

Please check the uploaded file of "response to reviewers".

---

## [Decision Letter · Decision Letter 1]

10 Sep 2023

The microRNA-mediated gene regulatory network in the hippocampus and hypothalamus of the aging mouse

PONE-D-23-13060R1

Dear Dr. Lee,

We’re pleased to inform you that your manuscript has been judged scientifically suitable for publication and will be formally accepted for publication once it meets all outstanding technical requirements.

Kind regards,

Tanja Grubić Kezele, Ph.D., M.D.

Academic Editor

PLOS ONE

Additional Editor Comments (optional):

Reviewers' comments:

Reviewer's Responses to Questions

**Comments to the Author**

1. If the authors have adequately addressed your comments raised in a previous round of review and you feel that this manuscript is now acceptable for publication, you may indicate that here to bypass the “Comments to the Author” section, enter your conflict of interest statement in the “Confidential to Editor” section, and submit your "Accept" recommendation.

Reviewer #2: All comments have been addressed

2. Is the manuscript technically sound, and do the data support the conclusions?

Reviewer #2: Yes

3. Has the statistical analysis been performed appropriately and rigorously? 

Reviewer #2: Yes

4. Have the authors made all data underlying the findings in their manuscript fully available?

Reviewer #2: Yes

5. Is the manuscript presented in an intelligible fashion and written in standard English?

Reviewer #2: Yes

6. Review Comments to the Author

Reviewer #2: The authors have now addressed my concerns and I believe the manuscript warrants publications at present.

7. PLOS authors have the option to publish the peer review history of their article (what does this mean?). If published, this will include your full peer review and any attached files.

Reviewer #2: No

---

## [Editor Report · Acceptance letter]

2 Nov 2023

PONE-D-23-13060R1 

The microRNA-mediated gene regulatory network in the hippocampus and hypothalamus of the aging mouse 

Dear Dr. Lee:

I'm pleased to inform you that your manuscript has been deemed suitable for publication in PLOS ONE. Congratulations! Your manuscript is now with our production department. 

Kind regards, 

on behalf of

Prof. dr. Tanja Grubić Kezele 

Academic Editor

PLOS ONE